# Brassinosteroids Induce Strong, Dose-Dependent Inhibition of Etiolated Pea Seedling Growth Correlated with Ethylene Production

**DOI:** 10.3390/biom9120849

**Published:** 2019-12-09

**Authors:** Petra Jiroutová, Jaromír Mikulík, Ondřej Novák, Miroslav Strnad, Jana Oklestkova

**Affiliations:** Laboratory of Growth Regulators, Institute of Experimental Botany, The Czech Academy of Sciences, & Faculty of Science, Palacký University, Šlechtitelů 27, 78371 Olomouc, Czech Republic

**Keywords:** brassinosteroid, growth inhibition, bioassay, Pisum sativum (var. arvense) sort. arvica, ethylene, 1-aminocyclopropane-1-carboxylic acid

## Abstract

We have recently discovered that brassinosteroids (BRs) can inhibit the growth of etiolated pea seedlings dose-dependently in a similar manner to the ‘triple response’ induced by ethylene. We demonstrate here that the growth inhibition of etiolated pea shoots strongly correlates with increases in ethylene production, which also responds dose-dependently to applied BRs. We assessed the biological activities of two natural BRs on pea seedlings, which are excellent material as they grow rapidly, and respond both linearly and uni-phasically to applied BRs. We then compared the BRs’ inhibitory effects on growth, and induction of ethylene and ACC (1-aminocyclopropane-1-carboxylic acid) production, to those of representatives of other phytohormone classes (cytokinins, auxins, and gibberellins). Auxin induced ca. 50-fold weaker responses in etiolated pea seedlings than brassinolide, and the other phytohormones induced much weaker (or opposite) responses. Following the optimization of conditions for determining ethylene production after BR treatment, we found a positive correlation between BR bioactivity and ethylene production. Finally, we optimized conditions for pea growth responses and developed a new, highly sensitive, and convenient bioassay for BR activity.

## 1. Introduction

Brassinosteroids (BRs) are a group of naturally occurring phytohormones with characteristic steroidal structure. BRs are essential for plenty of developmental and physiological processes such as cell elongation, cell division, leaf senescence, vascular differentiation, flowering time control, male reproduction, photomorphogenesis, and responses to both biotic and abiotic stresses [1,2,3]. Based on this, BRs are considered potent plant growth regulators and have been used to enhance the growth and yields of important agricultural crops [4]. Since BRs are present in plants in extremely low concentrations and have potent biological activities, their identification requires highly sensitive bioassays, based on responses to BRs that are not influenced by other endogenous plant hormones. 

Ethylene, the gaseous plant hormone with a very simple structure consisting of two carbon and four hydrogen atoms, is produced in most plant tissues and cell types. Crucial processes in plants, such as seed germination, growth, apical hook formation, organ senescence, fruit ripening, abscission, gravitropism, and stress responses, are affected by this hormone [5,6]. The most known effect of ethylene on etiolated seedlings is called a ‘triple response’, which consists of inhibition of stem elongation, radial swelling of the stem, and impairment of the normal geotropic response (formation of an exaggerated apical hook). This seedling phenotype has been used for identifying ethylene-related mutants [7,8]. Ethylene biosynthesis involves three main steps. Firstly, the amino acid methionine is converted to S-adenosyl-methionine (SAM); this step is catalyzed by the specific enzyme SAM synthetase (SAMS). The next (generally rate-limiting step) in ethylene biosynthesis is the conversion of S-adenosyl-methionine (SAM) to 1-Aminocyclopropane-1-carboxylic acid (ACC) catalyzed by 1-Aminocyclopropane-1-carboxylic acid synthase (ACS). finally, ACC is converted to ethylene, catalyzed by ACC oxidase (ACO) [9]. Interestingly, Tsang et al. [10] suggest that ACC, the direct precursor of ethylene, can act as an active signaling molecule itself, independent of ethylene production.

Several studies have shown that BRs stimulate ethylene production in various plant tissues [11,12,13]. One of the main mechanisms of how BRs could positively influence ethylene biosynthesis is via stabilization of ACC synthase the crucial enzyme in ethylene biosynthesis [14]. However, in a recent study of BRs’ effects on root growth, Lv et al. [15] found that they can have either of two effects on ethylene synthesis in Arabidopsis roots, depending on the applied concentration. Ethylene production was greatly reduced in seedlings treated with a low concentration (10 or 100 nM) of 24-epibrassinolide (24-epiBL), while a higher concentration (≥ 500 nM) strongly enhanced ethylene production. Chromatin immunoprecipitation (ChIP)/qPCR analysis showed that interactions of BR-regulated transcription factors BES1 (BRI1-EMS-SUPPRESSOR1) and BZR1 (BRASSINAZOLE-RESISTANT 1) with the promoter of ACSs, play important roles in these responses. The interactions are inhibitory because the expression of ACS is strongly suppressed when the BR transcription factors are over-expressed, and vice versa, ACS expression is increased in BR-insensitive mutants. Altogether these results suggest that at physiological levels, BRs repress ethylene biosynthesis via interaction with BES1 and BZR1 transcription factors and the promoters of ACSs, encoding the key ethylene biosynthetic enzyme, while at high levels, BRs and auxins synergistically induce ethylene production in Arabidopsis roots [15]. We recently discovered that brassinolide (BL) application has strong effects on etiolated pea seedlings, including all three phenotypic elements of the classical ‘triple response’ to ethylene (elongation and radial swelling of the hypocotyl, and exaggerated apical hook formation). Thus, in the study presented here, we tested the hypothesis that BRs’ biological activities may be mediated by ethylene, and the specificity of their activities, by examining corresponding activities of other plant hormones. We also developed a robust, sensitive, and convenient bioassay, the pea seedling growth inhibition test (in which ethylene production could also be monitored), for evaluating hormonal activities of new synthetic BR derivatives with potential agricultural uses. 

## 2. Results and Discussion

### 2.1. Effects of Brassinosteroids on Growth of Etiolated Pea Seedlings

First, we analyzed effects of two exogenously applied BRs (BL and 24-epiBL) at various concentrations on the growth of etiolated pea seedlings and found that BRs change their growth pattern. After treatment with brassinosteroids at higher concentration than 2 µM, we observed a reduced rate of elongation (Figure 1a,b). This effect is also accompanied by adeclining weight of epicotyls biomass (Figure 1d). Besides the inhibition and losing biomass of epicotyls, we also observed in these plants increased lateral expansion (Figure 1c), leading to swelling of the regions below the hook. It is evident from IC_50_ values (Table 1) that this response of pea hypocotyls is highly sensitive to BRs. The results with 24-epibrassinolide (BL IC_50_—2.2 × 10^−5^ M; 24-epiBL IC_50_—1.86 × 10^−5^ M) showed that it is a bit less active than brassinolide. Inhibitory effects of BRs on hypocotyl elongation of dark-grown plants have also been observed by Tanaka et al. [16], who found that BL inhibited the elongation of etiolated Arabidopsis plants’ hypocotyls at concentrations higher than 0.01 µM. In addition to inhibiting the growth and inducing swelling of etiolated pea seedlings, BRs also caused curvature of their etiolated stems, leading to an exaggerated apical hook (Figure 1a). These are three phenotypic elements of the typical ‘triple response’ of etiolated plants to ethylene observed in most dicots, including Arabidopsis [8]. Therefore, we examined BRs’ effects on ethylene production in the seedlings.

### 2.2. Inhibitory Effects of Other Plant Growth Regulators on Epicotyl Growth

To gauge the BR-specificity of the observed inhibitory effects on growth of etiolated plants, we tested effects of exogenous applications of representatives of the other main phytohormonal groups: auxin (indole-3-acetic acid, IAA), gibberellin (gibberellic acid, GA3), and cytokinins (trans-zeatin riboside, tZR, and thidiazuron, TDZ). The structures of these compounds are shown in Figure 2. Auxin had much stronger inhibitory effects on pea hypocotyl growth and elongation than the gibberellin and cytokinins (Figure 3), but only at substantially higher concentrations than the BRs (IC_50_ values for IAA, BL, and 24-epiBL: 1.78 × 10^−3^ M, 2.2 × 10^−5^ M, and 1.86 × 10^−5^ M, respectively). Thus, ca. 80-fold more IAA than BL was required. The cytokinins also inhibited elongation, but their IC_50_ values were ca. 1000-times higher than those of the BRs. Similarly, Chory et al. [17] found that the natural cytokinin N^6^-isopentenyladenine inhibited hypocotyl elongation of etiolated Arabidopsis at a much higher concentration (3.10^−6^ M) than BRs. Finally, treatment of the plants with gibberellic acid (GA3) had the opposite effect, causing etiolated pea stems to lengthen, in accordance with findings by Cowling and Harberd [18] that 14-day-old Arabidopsis plants treated with 10^−6^ M GA4 had longer hypocotyls than non-treated controls. Data presented in Table 1 clearly show that the seedlings responded highly sensitively and dose-dependently to the applied BRs. As already mentioned, in addition to inhibiting growth, BRs caused swelling and curvature of the seedlings’ etiolated stems.

### 2.3. Effects of BRs and Other Phytohormones on Ethylene Production in Etiolated Pea Seedlings

The results presented above clearly indicate that the inhibitory effect of BRs is mediated by endogenous ethylene biosynthesis. Thus, we determined ethylene production using a method that had been optimized with respect to treatment duration and temperature. Seedlings treated with a BR (or other phytohormone) at a given concentration are hermetically sealed in an Erlenmeyer flask, incubated in the dark at 22 °C and ethylene levels in the flask are measured after 24 h (when ethylene levels peaked in optimization tests; Figure 4). Note, however, Figure 4 shows that ethylene levels were higher after 24 h than after 12 and 6 h, but that they had not peaked then. The largest amounts of ethylene were produced by plants treated with 20 mM IAA or BRs (Figure 5), supporting the hypothesis that BRs’ inhibitory effects on etiolated pea seedlings are mediated by increases in ethylene production. Moreover, the minimum concentrations of BL (or 24-epiBL) and IAA required to elicit significant effects on ethylene production were ca. 20 nM and 20 µM, respectively. Thus, ethylene production in pea stems clearly responds much more sensitively to BRs (apps.100-times) than to IAA.

High ethylene production in plants treated with auxins is not surprising as auxin-induced ethylene production has been observed in numerous plant species [14,19,20]. BRs have also been shown to induce the production of ethylene, both alone and synergistically with other phytohormones in etiolated mung bean seedlings [12,21]. However, etiolated pea seedlings appear to be the most sensitive systems tested to date, responding detectably to as little as 100 fmol of BL. Mechanistic evidence that BR and auxin promote ethylene production has been provided by Joo et al. [22], who showed that 24-epiBL induces expression of the auxin-responsive ACC synthase gen AtACS4 in Arabidopsis. In addition, the cross-talk of BRs with ethylene is important for germination of seeds under salinity stress [23]. All this published information is consistent with our observations that BRs inhibit growth of pea seedlings’ stems and promote ethylene production in them. 

The application of TDZ also induced an increase in ethylene production, but only at the strongest (very high) concentration used (20 mM). These results are consistent with demonstrations that TDZ promotes ethylene evolution in several plant species [24,25], and is used for this purpose in cotton defoliation. Similarly, Lorteau et al. [26] found that the cytokinin 6-benzylaminopurine (BAP) stimulated ethylene production in pea roots (ethylene production was measured 6 h after the cytokinin treatment) The time between the administration of cytokinin and the ethylene determination appears to be decisive for the final amount of ethylene measured. James Rushing’s work [27] shows that ethylene production in broccoli florets treated by 6-benzylaminopurine (BAP) or zeatin peaked on the 2nd day, and dropped to control levels after four days. In stark contrast, ethylene production was considerably lower in our seedlings treated with tZR. High doses of GA3 cause a slight increase in ethylene production, but these levels are insignificant compared to the effect of BRs, IAA, and TDZ (Figure 5). Many studies have shown that ethylene can modulate gibberellin action or concentration [28,29,30], but the reverse interaction has received much less attention. However, Ferguson et al. [31] found that GA1 can probably suppress ethylene production because GA1-deficient pea mutants produced nearly twice as much ethylene as wild-type plants.

### 2.4. Effect of ACC Treatment on Epicotyl Growth and Ethylene Production in Etiolated Pea Plants

To verify that the ‘triple response’ of pea seedlings after BR treatment is caused by an increase in ethylene production, we treated etiolated pea seedlings with direct ethylene precursor 1-aminocyclopropane-1-carboxylic acid (ACC). The treatment with the highest tested concentration of ACC (20 mM) caused both inhibition of etiolated growth and increased level of ethylene production (Figure 6). Those results support the assumption that BR induces ethylene production, which leads to the ‘triple response’ of etiolated pea seedlings. 

### 2.5. Determination of ACC, a Direct Biosynthetic Precursor of Ethylene in Plants Treated with BRs in Time

As already mentioned, there are strong indications that BRs promote ethylene biosynthesis in seedlings by stimulating transcription of ACS genes and increasing the stability of ACS5 and ACS9 proteins [32]. Alternatively, BRs may suppress ethylene biosynthesis through interaction with BES1 and BZR1 transcription factors and the promoters of ACSs genes, encoding the key ethylene biosynthetic enzyme at BR levels below some threshold, but at higher levels induce ethylene production in conjunction with auxins [15]. To elucidate whether the increased ethylene production we observed after BR treatment was due to an increases in ACC biosynthesis, we measured the time courses of changes in concentrations of ACC and ethylene in BR (24-epiBL)-treated pea seedlings. As shown in Figure 7, ethylene production increased over time and peaked 36 h after the treatment, in accordance with previous findings that BRs may enhance ethylene production in etiolated plants treated with BR at times ranging from 8 h [33] to 3 d [34]. ACC levels in 24-epiBL-treated plants also peaked 36 h after treatment, and strongly correlated with ethylene production. These data corroborate the finding by Hansen et al. (2009) that induction of ethylene production by BR treatment is strongly linked to ACC biosynthesis.

### 2.6. Development of a New Bioassay

Several bioassays for BRs have been developed. In the past the two most commonly used are the bean second internode elongation (BSIE) assay and rice leaf lamina inclination test (RLIT). In the BSIE assay, elongation of the second internode of bean (Phaseolus vulgaris) seedlings is recorded. This elongation is characteristically accompanied by curvature, swelling, and splitting; the effects sometimes referred to as ‘the brassin response’. In this bioassay, auxins are inactive and gibberellins only cause elongation of the treated and upper internodes [35]. In the RLIT, explants (each consisting of leaf lamina, lamina joint, and leaf sheath) are excised from etiolated rice seedlings and floated on test solutions, then the inclination angle induced by test compounds is recorded [36]. In a modified version of the RLIT, intact dwarf rice (Oryza sativa) seedlings are used, and a test solution is applied as a micro drop at the junction between the lamina and the sheath. In the RLIT auxins are active, but at much higher concentrations than BRs. Gibberellins induce a straight growth response without bending of the leaf. Another assay is based on the fluorometric measurement of nitric oxide production by tomato suspension-cultures, which is induced by BL [37].

Based on the data presented in the previous sections, we developed a new bioassay, ‘the pea growth inhibition biotest’, for testing BRs’ biological activity. This biotest (Figure 9) is one of the most sensitive BR assays because as little as 100 fmol of BL can induce the monitored responses (Table 2). The elongation of the stems is linearly dependent on the logarithm of BL concentration over four orders of magnitude (Figure 8), and inter-assay variability is about 8%. We found that several factors affect this biotest’s sensitivity. Firstly light: as etiolated plants are used, it is essential to perform all operations in the dark or in green light (540 nm). Another important factor is the application of BRs to the plants in droplets of fractionated lanolin (Figure 9), because the BRs must be in continual contact with the plants’ tissues. The sensitivity is also dependent on the pea cultivar. We compared responses of numerous cultivars and found that Pisum sativum (var. arvense) sort. Arvica is highly suitable because it grows rapidly and its elongation response to BRs is uni-phasic [39].

## 3. Materials and Methods

### 3.1. General Information

All chemicals and solvents were purchased commercially and used without further purification. Chemical compounds applied in this study were brassinolide, 24-epibrassinolide, indole-3-acetic acid, gibberellin GA3, trans-zeatin, thidiazuron, and [^2^H_4_] 1-aminocyclopropane-1-carboxylic acid (PubChem CID: 115196, 443055, 802, 9819600, 449093, 40087and 84392-07-4 respectively). All these compounds were obtained from Olchemim s.r.o. (Olomouc, Czech Republic). The experimental plants were etiolated pea Pisum sativum (var. arvense) sort. Arvica seedlings (seeds were obtained from MORSEVA s.r.o., Olomouc, Czech republic).

### 3.2. Pea Seedling Cultivation

Pea seeds were germinated for 2 d on moist filter paper in the dark, then uniform seedlings from a large population were transferred into pots containing perlite and 1/10 diluted Hoagland solution (half-concentration, pH 5.7). The pots were placed in a dark cultivation room (24 °C, relative humidity 75%), and 24 h later, the seedlings were treated with various amounts of test compounds in 5 µL fractionated lanolin. The substances were applied in micro drops to the scar left after bract removal (Figure 9). Control plants were treated with lanolin alone. The length of etiolated pea stems was measured after 4 d (Figure 9) and the difference in length between treated and control plants was used as a measure of activity. Sets of eight seedlings were subjected to each treatment (exposure to one of the test compounds at one of the concentrations) in each of three independent experiments, the *p*-values were calculated with a two-tailed Student *t*-test using Excel software (Microsoft Corporation, Redmond, WA, USA).

### 3.3. Determination of Ethylene Production

To measure ethylene production, pea seedlings (eight per treatment) were placed in 0.5 l glass containers for 24 h in the dark. A portion (1 mL) of headspace gas was withdrawn from each container by syringe for each measurement and injected into a GC System gas chromatograph equipped with a flame ionic detector (FID) and HP-AL/S capillary column (50 m × 0.535 mm × 15 μm), all from Agilent Technologies (Santa Clara, CA, USA). The chromatographic settings were: column temperature, 150 °C; detector temperature, 220 °C; carrier gas. The area under the resultant peak (*y-*axis) versus sensitivity (*x*-axis; nl.mL^−1^) was representing a quantitative measure of ethylene concentration, *p*-values were calculated with two-tailed Student *t*-test using Excel software (Microsoft Corporation, Redmond, WA, USA).

### 3.4. ACC determination

The tissue (50 mg of etiolated pea plants) was homogenized in 1 mL of H_2_O: methanol:chloroform (1:2:1), 50 pmol of internal standard ([D_4_]ACC) was added to each sample, and after centrifugation (4 °C, 15 000 rpm) the supernatant was collected and evaporated to dryness. The samples were derivatized using an AccQ-Tag Ultra kit (Waters, Milford, MA, USA) and subsequently analyzed by an ultra-performance liquid chromatography-tandem mass spectrometry (UPLC-MS/MS) system consisting of an ACQUITY UPLC^®^ I-Class system (Waters, Milford, MA, USA) and a Xevo^TM^ TQ-S MS triple quadrupole mass spectrometer (Waters MS Technologies, Manchester, UK) [40].

## 4. Conclusions

The etiolated plants treated with brassinosteroids in higher concentrations than 0.2 µM showed the declining weight of epicotyls biomass and increasing lateral expansion, leading to swelling of the regions bellow the hook. Because inhibited plants had signs of ‘triple response’ to ethylene, we also developed a method for ethylene measurement and examined its production together with its biosynthetic precursor ACC. Ethylene production increased with time after treatment and peaked in 36 h; these results correlate with ACC accumulation in these plants. Based on these results, a new sensitive bioassay that uses etiolated pea plants has been developed. The biotest is sensitive for BRs; as little as 100 fmol of BR can be detected.

## Figures and Tables

**Figure 1 biomolecules-09-00849-f001:**
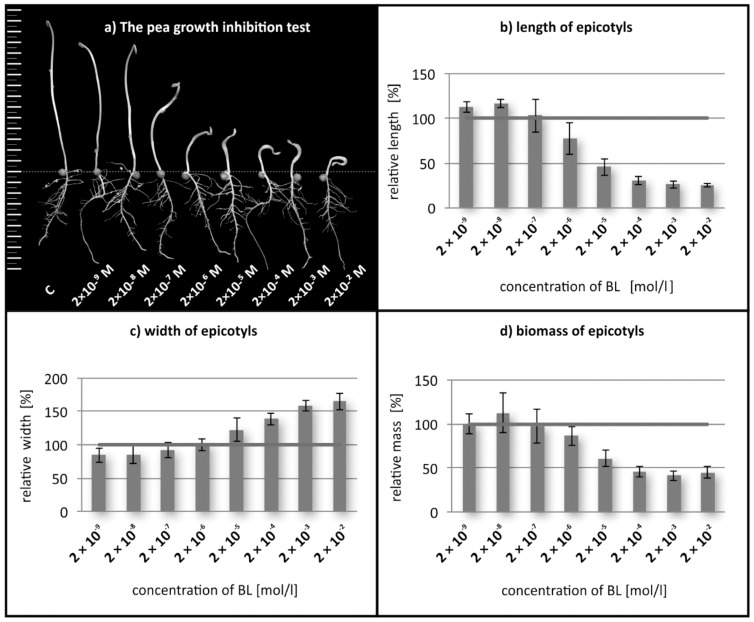
Visual effects of brassinolide (BL) on etiolated pea plants (**a**) and quantified effects on the length (**b**), width (**c**) and biomass (**d**) of epicotyls treated with BL at indicated concentrations. Error bars represent S.D.

**Figure 2 biomolecules-09-00849-f002:**
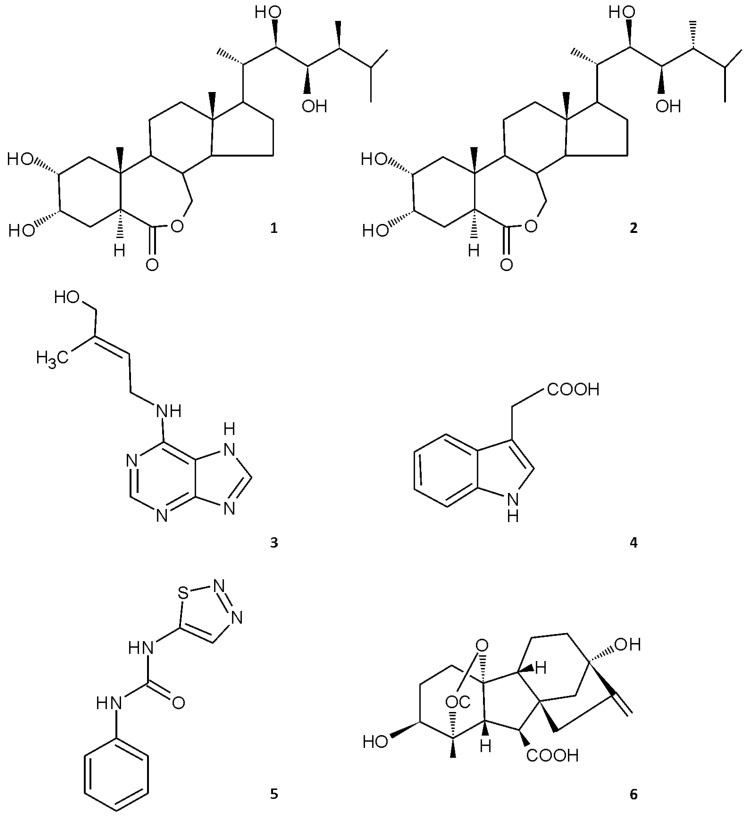
Structures of tested growth regulators: brassinolide (**1**), 24-epibrassinolide (**2**), trans-zeatin (**3**), indole-3-acetic acid (**4**), thidiazuron (**5**), gibberellic acid (GA3) (**6**).

**Figure 3 biomolecules-09-00849-f003:**
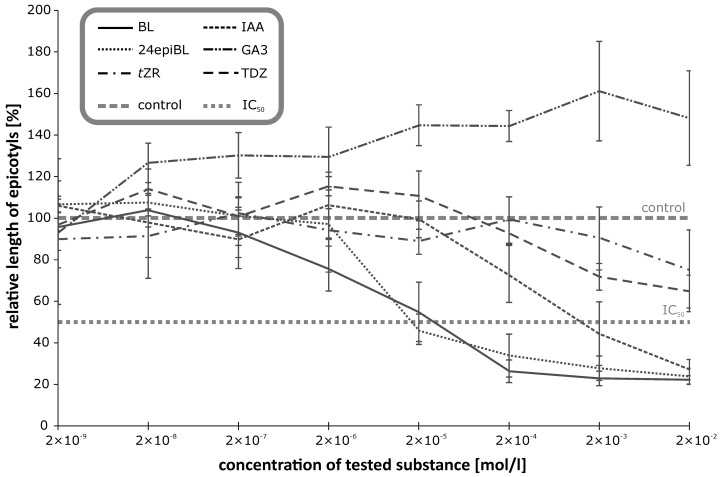
Effect of selected growth regulators on the inhibition of etiolated pea seedlings’ growth. Error bars represent standard deviations of the means. (For statistical data see Appendix A).

**Figure 4 biomolecules-09-00849-f004:**
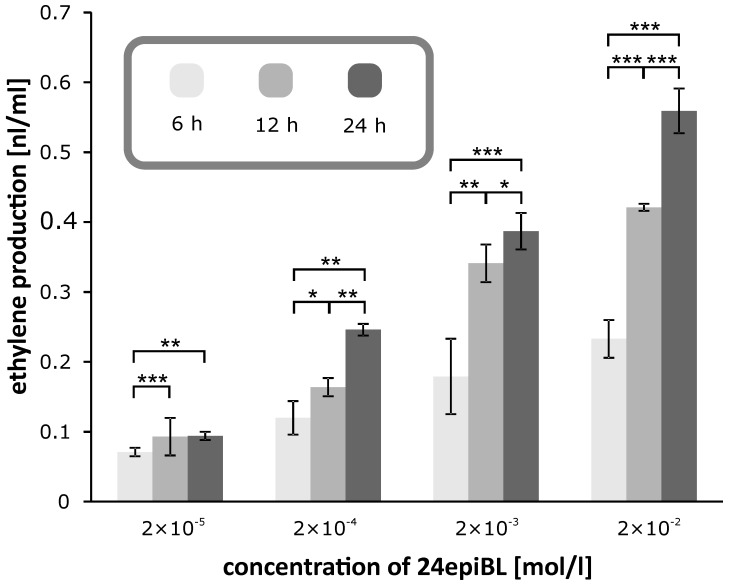
Effects of 24-epibrassinolide (24-epiBL) on ethylene production (nl/mL) by etiolated pea seedlings determined by GC-FID (gas chromatograph equipped with flame ionization detector) 6, 12 and 24 h after ventilation. Error bars represent standard deviations of the means. Error bars represent S.D. Asterisks represent significant changes (*t*-test), * represents *p* value < 0.05, ** represent *p* value < 0.01, *** represent *p* value < 0.001.

**Figure 5 biomolecules-09-00849-f005:**
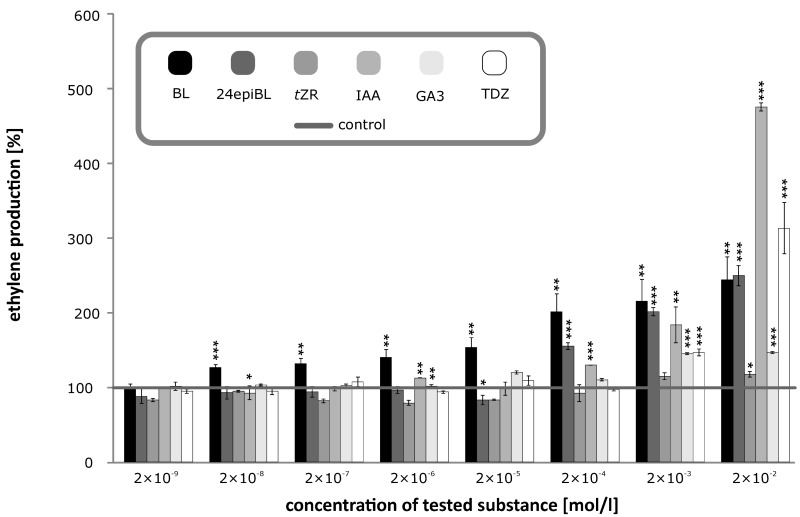
Effects of selected growth regulators on ethylene production (nl/mL) by etiolated pea seedlings determined by GC-FID 24 h after ventilation. Error bars represent standard deviations of the means. Error bars represent S.D. Asterisks represent significant changes (*t*-test), * represents *p* value < 0.05, ** represent *p* value < 0.01, *** represent *p* value < 0.001.

**Figure 6 biomolecules-09-00849-f006:**
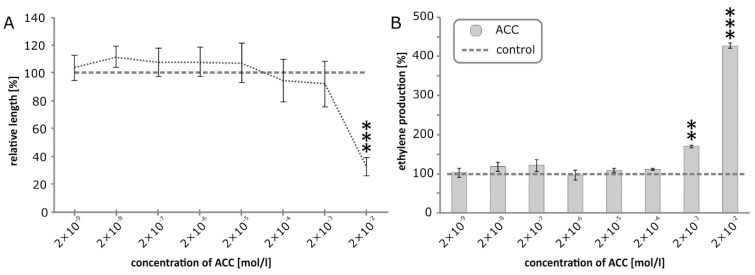
Effect of direct ethylene precursor 1-aminocyclopropane-1-carboxylic acid (ACC) on inhibition of etiolated pea seedlings’ growth (**A**) and on ethylene production by etiolated pea seedlings determined by GC-FID 24 h after ventilation (**B**). Error bars represent standard deviations of the means. Asterisks represent significant changes (*t*-test), ** represent *p* value <0.01, *** represent *p* value < 0.001.

**Figure 7 biomolecules-09-00849-f007:**
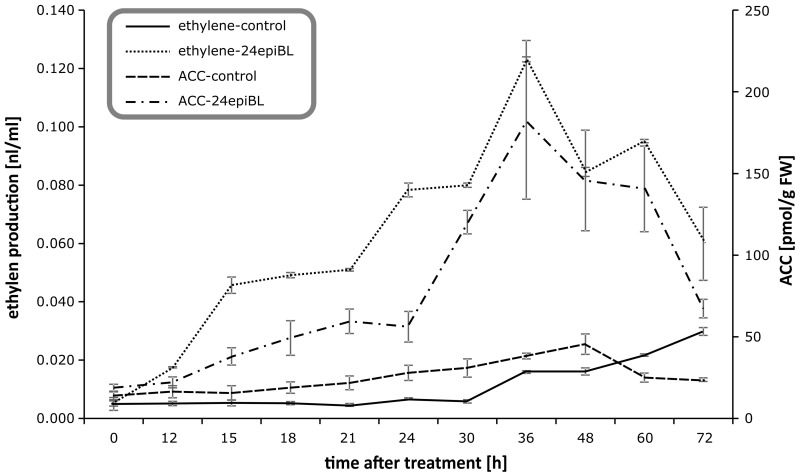
Effects of 24-epiBL on ethylene production (nl/mL) and concentration of 1-aminocyclopropane-1-carboxylic acid (ACC) (pmol/g FW) in etiolated pea seedlings determined by GC-FID (gas chromatograph equipped with flame ionization detector) resp. UPLC-MS/MS (ultra-performance liquid chromatography-tandem mass spectrometry). Error bars represent standard deviations of the means.

**Figure 8 biomolecules-09-00849-f008:**
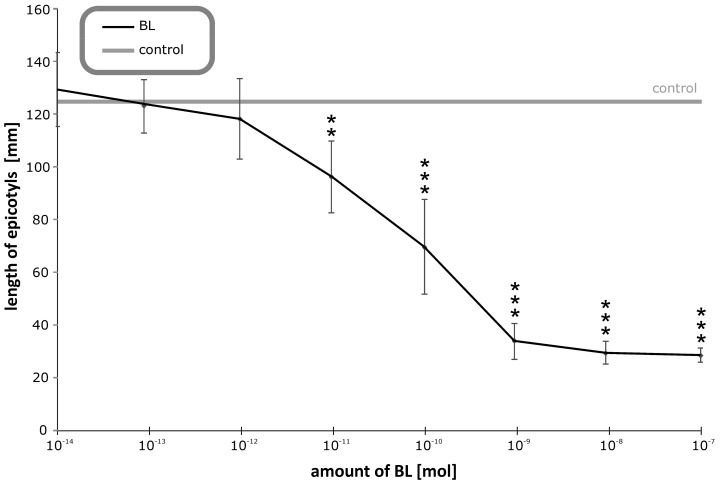
Inhibitory effect of brassinolide (BL) on etiolated pea seedlings’ growth. Error bars represent S.D. Asterisks represent significant changes (*t*-test), * represents *p* value < 0.05, ** represent *p* value < 0.01, *** represent *p* value < 0.001.

**Figure 9 biomolecules-09-00849-f009:**
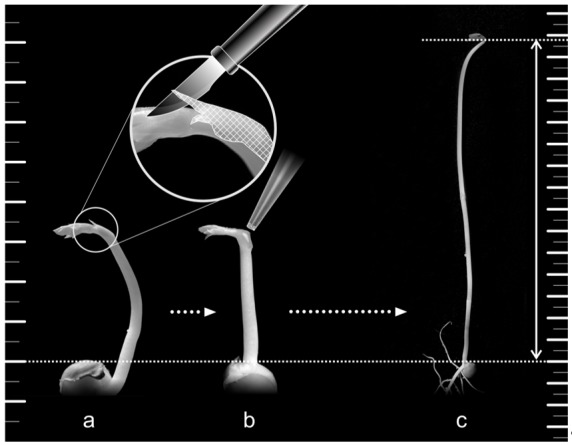
Scheme of the pea inhibition assay—cutting of bract (**a**), application of tested compound in micro drop of lanolin on the scar formed by bract removal (**b**) measurement of epicotyl length (**c**).

**Table 1 biomolecules-09-00849-t001:** IC_50_ (mol/L) values of selected brassinosteroids and other phytohormones obtained from the pea growth inhibition biotest, in which the etiolated growth of pea seedlings is inhibited.

IC_50_	Concentration [mol/L]
BL	2.2 × 10^−5^
24-epiBL	1.86 × 10^−5^
tZR	2.99 × 10^−2^
IAA	1.78 ×10^−3^
TDZ	2.59 × 10^−2^
GA3	no inhibition

**Table 2 biomolecules-09-00849-t002:** The sensitivity of the pea growth inhibition biotest and three previously described bioassays for brassinosteroids (BRs).

Bioassay	Detection Limit	Reference
BSIE	20 pmol	[38]
RLIT	0.1 pmol	[38]
NO production bioassay	0.5 pmol	[37]
Pea inhibition bioassay	0.1 pmol	This study

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
