# Peer review of "Brassinosteroids Induce Strong, Dose-Dependent Inhibition of Etiolated Pea Seedling Growth Correlated with Ethylene Production"

_biomolecules, 2019, doi:10.3390/biom9120849_

Round 1
Reviewer 1 Report
The authors have improved the manuscript and it is ready for publication. I do have one remaining concern. The authors maintain their argument that the assay is specific to BRs, but that is simply not the case and the conclusion must be withdrawn from the paper prior to publication. Even in the highly limited case of synthetic BR screens, an ethylene agonist would have the same phenotypes and therefore the assay is not specific. In reality, any compound that is growth inhibitory could slow hypocotyl elongation - and there are many such compounds beyond the few hormones tested - would result in a false positive in this assay. This is not a major concern with the assay as specificity is not required since follow up experiments could confirm BR activity easily enough. The specificity wording should simply be removed from the discussion.
Author Response
Thank you for your comments. According to your instructions we have removed the phrase “ highly specific” from the discussion and conclusin.
Reviewer 2 Report
I was satisfied with the present version. It can be published in Biomolecules.
Author Response
Thank you for your positive comments.
Reviewer 3 Report
This paper by Jiroutová et al has clearly benefited from the revision, as advised by reviewers. Readability is significantly enhanced and the text is more concise.
In the present work, the authors investigate the involvement of brassinosteroids in inhibition growth of etiolated pea seedling in correlation with ethylene production. This part is interesting and clearly provides new data valuable for the research community. The paper title is well stated, it is informative and concise. Abstract and introduction are well written. Material and research methods are presented appropriately and clearly. Experimental setup and the description in the methods section are well structured, precise enough, clearly described and the statistical analysis is done alright.
The results obtained in this study are interesting, and discussion of results is correct and sufficient.
Author Response
Thank you for your positive comments.
This manuscript is a resubmission of an earlier submission. The following is a list of the peer review reports and author responses from that submission.
Round 1
Reviewer 1 Report
The article "Brassinosteroids induce strong, dose-dependent inhibition of etiolated pea seedling correlated with ethylene production" describes a straightforward body of work characterising the response of etiolated pea seedlings to various hormones at various concentrations. The authors focus on brassinosteroids because of a relatively sensitive response involving ethylene production and resulting morphological changes in seedling development. The authors also argue that they can use their system as a sensitive BR bioassay. The hormone treatment results and bioassay idea are interesting, but as they have been shown in other species including other legumes, they are not particularly novel.
Major comments:
The authors argue that BR induces ethylene production and this explains the triple response, but they do not treat pea seedlings with ethylene directly to show the triple response to ethylene alone. This would improve the manuscript.
The ethylene treatment experiment is also important because the authors claim that their bioassay is specific, but if ethylene (or ACC) has the same effects, then the assay is not specific to BR.
The authors do not present any example of a proof of principle application of their bioassay to measure BR from biological samples. This would strengthen their claims of bioassay sensitivity. It would seem that IC50 of ~20 µM is not particularly sensitive for detection of biological BR samples.
The authors show that GA3 treatment increases ethylene, but then argue that GA1 suppression of ethylene production is in accordance with their results.
Minor comments:
1. Some sections require editing for english grammar and spelling (e.g. 3.1).
2. Line 172 apps.100-times ... should be 1000X?
3. Should keep concentration notations the same throughout (i.e. use either E-05 M or 10-5 M but not both).
4. Text in figure 1a is hard to read.
5. Title of Table I does not explain what is being inhibited.
Reviewer 2 Report
Dear authors,
The research of this manuscript was focus on the effects of exogenous BRs on the growth of pea seedlings. Then, the authors developed a new method for the measurement of BR in etiolated pea seedling. Moreover, the inhibitive effects of BRs to seedling was strongly consistent with the effects of ethylene. In the presence of the production of endogenous ethylene, ACC content was consistent with the content of ethylene. This is a valuable discovery. Revealed that the signal pathway might have an intersection between ethylene and BR. Based on this, I think this manuscript is worth to be published.
Reviewer 3 Report
In the present work, the authors investigate the involvement of brassinosteroids in inhibition growth of pea seedling in correlation with ethylene production. This part is novel and interesting and clearly provides new data valuable for the research community. TITLE The paper title is well stated, it is informative and concise. ABSTRACT, INTRODUCTION This section is well written. MATERIAL AND METHODS Material and research methods are presented appropriately and clearly. Experimental setup and the description in the methods section are well structured, precise enough, clearly described and the statistical analysis is done alright. RESULTS The results obtained in this study are interesting. DISCUSSION In general, the discussion of results is correct and sufficient. LITERATURE The items of literature included in the paper are rather sufficient and adequate to the subject of the paper.Reviewer 4 Report
The ms by Jiroutova et al describe the seemingly counterintuitive inhibitory effect of high doses of BRs on the elongation of seedlings, and adscribe this effect to the accumulation of ethylene triggered by BRs. Finally, the authors put forward a sensitive bioassay in peas for BRs or its analogs.
My main concern with this ms is the message the authors intend to convey. On the one hand, the correlation between BRs application and ethylene production may be assumed as a causal relationship considering the litterature on the subject, but not necessarily from what the authors show in the ms. In fact, in my opinion the ms falls short as a classical plant physiology paper on the interaction between growth regulators; the authors do not perform, for example, treatments with inhibitors of ethylene production to check if they prevent the swelling and reduction in growth rates of epicotyls. On the other hand, the pea-based bioassay may indeed be more sensitive than others that have been used to assess the effects of BRs, but the conditions for its implementation may hinder its acceptance (dark treatments, the use of a particular cultivar of peas, and this only for epicotyl elongation: ethylene production is in fact more responsive to lower doses of BRs, but that would require specialized equipment).
On the formal side, the ms would benefit from a major reordering. The description of the experiments and the corresponding figures fails to flow swiftly. In section 3.1 the authors summarize the effects of different hormones (but significantly not ethylene or any precursor) on the growth of epicotyls in table 1, but the last sentence of this section links it to paragraph 3.3, not to the ensuing section 3.2, which instead comes back to the effect of the hormones already described; in fact, data in table 1 come from figure 3 (by the way, figure 2 doesn’t add much to the ms, we could assume that everyone is acquainted with the structure of the most common plant hormones or, at least, that’s an information easily retrievable). In the first sentence of section 3.3 the authors set out saying that “results presented… clearly indicate that the inhibitory effect of BRs is mediated by… ethylene…”, but they only address this question in the following sentences. Finally, the first figure one encounters in the ms is in fact the the last one (figure 8). The ms would also benefit from a few minor adjustments in the English expression and typos (for instance, in lines 118-119 “besides” instead of “except”, “increased” instead of “increasing” and “below” in place of “bellow”).
In Figure 1b, I don’t see that the effect of 2x10-7 M BRs is significantly different from the control.